# Suppressive Effect of Fraxetin on Adipogenesis and Reactive Oxygen Species Production in 3T3-L1 Cells by Regulating MAPK Signaling Pathways

**DOI:** 10.3390/antiox11101893

**Published:** 2022-09-24

**Authors:** Woonghee Lee, Gwonhwa Song, Hyocheol Bae

**Affiliations:** 1Institute of Animal Molecular Biotechnology, Department of Biotechnology, College of Life Sciences and Biotechnology, Korea University, Seoul 02841, Korea; 2Department of Oriental Medicinal Biotechnology, College of Life Sciences, Kyung Hee University, Yongin 17104, Korea

**Keywords:** fraxetin, obesity, adipogenesis, mitogen-activated protein kinase, ROS

## Abstract

Recent studies have identified obesity as one of the world’s most serious chronic disorders. Adipogenesis, in which preadipocytes are differentiated into mature adipocytes, has a decisive role in establishing the number of adipocytes and determining the lipid storage capacity of adipose tissue and fat mass in adults. Fat accumulation in obesity is implicated with elevated oxidative stress in adipocytes induced by reactive oxygen species (ROS). Adipogenesis regulation by inhibiting adipogenic differentiation and ROS production has been selected as the strategy to treat obesity. The conventional anti-obesity drugs allowed by the U.S. Food and Drug Administration have severe adverse effects. Therefore, various natural products have been developed as a solution for obesity, suppressing adipogenic differentiation. Fraxetin is a major component extracted from the stem barks of *Fraxinus rhynchophylla*, with various bioactivities, including anti-inflammatory, anticancer, antioxidant, and antibacterial functions. However, the effect of fraxetin on adipogenesis is still not clearly understood. We studied the pharmacological functions of fraxetin in suppressing lipid accumulation and its underlying molecular mechanisms involving 3T3-L1 preadipocytes. Moreover, increased ROS production induced by a mixture of insulin, dexamethasone, and 3-isobutylmethylxanthine (MDI) in 3T3-L1 was attenuated by fraxetin during adipogenesis. These effects were regulated by mitogen-activated protein kinase (MAPK) signaling pathways. Therefore, our findings imply that fraxetin possesses inhibitory roles in adipogenesis and can be a potential anti-obesity drug.

## 1. Introduction

A report published by the World Health Organization in 2021 states that four out of ten adults are overweight, and approximately 13% are obese globally [1]. Obesity has recently been acknowledged as one of modern society’s most urgent medical issues because it can be a decisive hazardous factor for various metabolic disorders, including hyperlipidemia, cardiovascular disease, and diabetes mellitus [2,3]. Adipogenesis is a process of multiplication and differentiation from preadipocytes into mature adipocytes [4]. Adipogenesis is strictly governed by several adipogenic transcriptional factors, including CCAAT-enhancer-binding protein α (C/EBPα) and peroxisome proliferator-activated receptor γ (PPARγ), which are essential for the expression of adipogenesis-related genes [5]. Activating these adipogenic genes is decisive in establishing the number of adipocytes, mostly developed before adulthood, and finally determining the lipid storage capacity of adipose tissue and fat mass in adults [6]. To overcome obesity, various strategies have been suggested, including lowering energy/food intake, increasing energy expenditure, reducing the differentiation of preadipocytes, suppressing lipogenesis, and stimulating lipolysis and fat oxidation [7]. Among these strategies, controlling adipogenesis by suppressing adipogenic differentiation has been commonly regarded as a valid approach to treating obesity [8].

Several medicines have been allowed by the U.S. Food and Drug Administration (FDA) as anti-obesity drugs [9]. Among them, orlistat is one of the most prescribed drugs, which acts as a lipase inhibitor, preventing the absorption of fats from consumed food [10]. However, orlistat has several serious gastrointestinal adverse effects, including steatorrhea, cholelithiasis, cholestatic hepatitis, and subacute liver failure [11]. In addition, other drugs that the FDA has approved also have limitations such as congenital disabilities (phentermine-topiramate), an increased risk of suicide (naltrexone-bupropion), pancreatitis (liraglutide), and abdominal pain (semaglutide) [9]. Thus, many reports have recently suggested that various natural products can act as anti-obesity medications by suppressing adipogenic differentiation [12,13]. Although many pharmacological drugs involving natural ingredients have been proposed to treat obesity, the likelihood of their clinical application is low due to an inadequate pharmacological approach and a lack of an etiologic–mechanistic perspective on obesity [14].

During adipogenesis, reactive oxygen species (ROS) stimulate adipocyte differentiation at an early stage and maturation at a late stage [15]. It has been reported that fat accumulation in patients with obesity is accompanied by elevated oxidative stress-derived ROS [16]. ROS may be hazardous for patients with diabetes by inhibiting glucose uptake in muscles and fat and the impending release of insulin from pancreatic cells [17,18]. Thus, the elevation of ROS neutralization can be one of the strategies for suppressing adipogenesis [19].

Fraxetin (6-methoxy-7,8-dihydroxycoumarin) is a major component extracted from the stem barks of *Fraxinus rhynchophylla*, which has been widely used in traditional medicine. Recent evidence suggests that fraxetin has versatile bioactivities, including anticancer, antioxidant, anti-inflammatory, and antibacterial properties [20,21,22,23]. Moreover, fraxetin is easily available, relatively inexpensive, and has few adverse effects and low resistance [23]. A study has shown that fraxetin exhibited an anti-obesity effect [24]. However, the effect of fraxetin on adipogenesis, especially its molecular mechanism and signaling pathway in 3T3-L1 cells, is still not clearly understood. The present study aims to verify the anti-obesity properties using 3T3-L1 cells, with respect to regulating lipid accumulation and its underlying molecular mode of action.

## 2. Materials and Methods

### 2.1. Chemicals

Fraxetin (Cat No. 18224) and orlistat (Cat No. O4139) were obtained from Sigma (St. Louis, MO, USA) and dissolved in dimethyl sulfoxide (DMSO). Table 1 shows the antibodies and various mitogen-activated protein kinase (MAPK) signaling inhibitors we used in the present study.

### 2.2. Cell Culture and Differentiation of Preadipocytes

The mouse preadipocytes 3T3-L1 were obtained from ATCC (Rockville, MD, USA). The 3T3-L1 cells were maintained in DMEM containing 10% bovine calf serum and 1% penicillin–streptomycin at 37 °C in a humid 5% CO_2_ atmosphere. The cells were subcultured when they reached 70% confluency. For the differentiation of preadipocyte to mature adipocyte, fully confluent 3T3-L1 cells (day 0) were incubated with a mixture of 0.5 mM 3-isobutylmethylxanthine (IBMX), 5 µg/mL insulin, and 1 µM dexamethasone (MDI) in DMEM supplemented with 10% fetal bovine serum (FBS) for 2 days. On day 2, the MDI medium was replaced with DMEM containing 10% FBS and 5 µg/mL insulin. On day 4, this medium was switched to DMEM containing 10% FBS. This medium was changed every two days until day 8. A diverse dose of fraxetin and other reagents was employed from day 0 to day 2. All experiments were performed in triplicate.

### 2.3. Cell Proliferation Test

We assessed cell proliferation of 3T3-L1 preadipocytes using Cell Proliferation Kit I (Cat No. 11465007001; Roche Holdings AG, Basel, Switzerland), which is also known as 3-(4,5-dimethylthiazol-2-yl)-2,5-diphenyl-2H-tetrazolium bromide (MTT) assay. The cells in 96-well culture plates were incubated with various concentrations of fraxetin and 10 µL of MTT tetrazolium salt at 37 °C for 4 h, followed by incubation in solubilization buffer at 37 °C overnight. Absorbance was assessed at 560 nm and 650 nm.

### 2.4. Oil Red O Staining

On day 8, from the onset of 3T3-L1 differentiation, the differentiated cells were washed with phosphate-buffered saline (PBS) and fixed with 4% formaldehyde for 1 h at room temperature. After being gently rinsed with 60% isopropanol, the cells were stained with filtered oil red O solution (Cat No. O1391, Sigma, St. Louis, MO, USA) for 30 min. The oil red O solution was removed and washed with distilled water four times. Images of lipid droplets in mature adipocytes were observed using a DM3000 microscope (Leica Microsystems GmbH, Wetzlar, Germany). The dye contained in lipid droplets was eluted with 100% isopropanol and quantified by assessing the absorbance at 490 nm.

### 2.5. Apoptosis Analysis

To analyze an apoptotic effect of fraxetin in 3T3-L1 cells, Annexin V and propidium iodide (PI) staining was performed using Annexin V apoptosis detection kit I (BD Bioscience, Franklin Lakes, NJ, USA). The cells were treated with different concentrations of fraxetin (0, 20, 50, and 100 µM) in an MDI induction medium for 48 h. The cells were rinsed with PBS twice and stained with Annexin V and PI at room temperature for 15 min. The fluorescence intensity was measured with a flow cytometer (BD Accuri C6, BD Bioscience). The experiment was repeated three times. The number of cells in each dot plot was 10,000.

### 2.6. Cell Cycle Assay

The alteration of cell cycle distribution was assessed using flow cytometry. Confluent preadipocytes were incubated with various concentrations of fraxetin-containing MDI for 2 days. Cells were collected and rinsed with PBS twice, followed by fixation with chilled 70% ethanol. Cells were stained by PI-containing ribonuclease A (Cat No. R5503, Sigma, St. Louis, MO, USA) at room temperature for 1 h. The fluorescence intensity was measured with a flow cytometer (BD Accuri C6, BD Bioscience, Franklin Lakes, NJ, USA). The experiment was repeated three times.

### 2.7. Determination of ROS

We evaluated total ROS products in preadipocytes and mature adipocytes stained with 2,7-dichlorofluorescin diacetate (DCFH-DA; Cat. No.: D6883, Sigma, St. Louis, MO, USA), which is transformed into fluorescent 2,7-dichlorofluorescein (DCF) in the presence of ROS. Cells were incubated with different doses of fraxetin for 1 h and stained with DCFH-DA for 30 min. Supernatants were collected and rinsed with PBS. The DCF intensity based on 10,000 cells per gate was calculated using the flow cytometer. The assay was performed in triplicate.

### 2.8. Western Blotting

The whole cell protein was isolated from preadipocytes and mature adipocytes, followed by immunoblotting, as described in our previous study [25].

### 2.9. RT-qPCR

Total RNA was isolated from preadipocytes and mature adipocytes, and transcriptional expression was determined as described in our previous study [25]. The detailed information on primers is represented in Table 2.

### 2.10. Statistical Analysis

All data were assessed with unpaired *t*-tests or analysis of variance, followed by Dunnett’s post hoc test using GraphPad Prism 7.00 (La Jolla, CA, USA). All data were obtained in triplicate. Values with *p* < 0.05 were considered statistically significant. Data are presented as means ± standard deviation.

## 3. Results

### 3.1. Impact of Fraxetin on the Formation of Lipid Droplets and the Cell Proliferation of 3T3-L1

Confluent 3T3-L1 was incubated with different concentrations of fraxetin during MDI induction to identify the influence of fraxetin on adipogenesis. On day 8, from the onset of differentiation, the accumulated lipids were stained with oil red O solution. Lipid accumulation was gradually inhibited by fraxetin compared with 0 µM (MDI)-treated cells (Figure 1A). Orlistat, a lipase inhibitor used for treating obesity [26], was used as a positive control. Compared with MDI-treated cells, the 10, 20, and 50 µM fraxetin treatments suppressed lipid accumulation by 81%, 68% *(p *< 0.05), and 74% *(p *< 0.05), respectively. It was noteworthy that the 100 µM fraxetin treatment inhibited lipid accumulation by 62% *(p *< 0.01), which was comparable with the 20 µM orlistat treatment (56%) *(p *< 0.01) (Figure 1B). The treatment of fraxetin did not show cytotoxicity until 100 µM (Figure 1C). However, 150 µM of fraxetin exhibited a significant cytotoxic effect in 3T3-L1 cells; therefore, we set the optimal dose as 100 µM in our study. Our observations suggest that fraxetin considerably suppressed lipid droplets’ formation in adipocytes.

### 3.2. Effect of Fraxetin on the Activity of Differentiation-Associated Proteins and Genes 

Because PPARγ and C/EBPα are known and essential players in the transcriptional network regulating the conversion from preadipocyte to adipocyte [27], we confirmed their protein expressions through immunoblotting (Figure 2A). In response to 20 µM and 100 µM of fraxetin, the protein levels of C/EBPα were significantly reduced to 0.64-fold *(p *< 0.05) and 0.52-fold *(p *< 0.01), respectively, compared to the differentiated cells. Moreover, the protein level of PPARγ also considerably decreased to 0.40-fold *(p *< 0.01) in response to 50 µM of fraxetin (Figure 2B). Consistent with the Western blot results, RT-qPCR data indicate that the transcriptional expressions of PPARγ and C/EBPα were also considerably decreased by 100 µM of fraxetin compared to the MDI induction (Figure 2C). Compared with the MDI-treated groups, the transcriptional activity of adipocyte-specific markers, including Fabp4, Fasn, and Srebf1 mRNA, was also significantly decreased by the treatment of fraxetin (Figure 2C). These results imply that fraxetin inhibited the differentiation of 3T3-L1 through the modulation of adipogenesis-related genes.

### 3.3. Fraxetin Inhibits Adipogenesis in the Early Stage of Differentiation

Confluent preadipocytes were incubated with 100 µM of fraxetin at various stages of differentiation, including early, intermediate, and late stages, to determine how fraxetin suppresses adipogenesis (Figure 3A). The formation rate of lipids was evaluated with oil red O. Compared to only MDI-treated control cells, the 6-day treatment of fraxetin (100 µM) significantly reduced intracellular lipid droplets by 54% *(p *< 0.001). The relative lipid droplets stained by oil red O solution were lower in fraxetin-treated cells in the early stage of differentiation than those in other stages (Figure 3B). Figure 3C shows that the inhibitory effect of fraxetin on lipid accumulation during the early, intermediate, and late stages was approximately 36% *(p *< 0.001), 15% *(p *< 0.01), and 11%, respectively. Our observation indicates that fraxetin inhibits adipogenesis most effectively during the early stage of adipogenesis. It was further evaluated by analyzing the transcriptional level of Kruppel-like factor 5 (Klf5), which is activated in the early stages of adipogenesis (Figure 3D). Our observations imply that fraxetin regulates the early stage of adipogenesis.

### 3.4. Fraxetin Alters Cell Distribution in the Cell Cycle without Apoptosis in 3T3-L1 Cells

We determined whether fraxetin influences cell distribution in the cell cycle. The 3T3-L1 preadipocytes were exposed to an MDI induction medium containing fraxetin for 48 h (Figure 4A). Compared to the G0/G1 phase (54%) and G2/M phase (43%) of nondifferentiated (ND) cells, MDI-treated cells in G0/G1 phase were reduced to 42% and increased in the G2/M phase to 55% (both* p *< 0.05). However, the changes were gradually reinstated by the treatment of fraxetin. After the treatment with fraxetin, the ratio of the G0/G1 phase gradually increased, and the G2/M phase gradually decreased. Compared to the MDI-treated cells, the G2/M-phase cells were reduced from 55 to 46% *(p *< 0.05) after 100 µM fraxetin treatment.

The manipulation of cell cycle progression is involved in the apoptotic response in multicellular organisms [28]. Therefore, we conducted Annexin V and PI staining to understand whether the changes in cell cycle progression induced by fraxetin affect the apoptotic response in 3T3-L1 cells. Figure 4B shows that fraxetin did not change the number of apoptotic cells. Our findings imply that fraxetin exhibits inhibitory effects on adipogenesis by the alteration of cell distribution in the cell cycle without apoptosis.

### 3.5. Fraxetin Attenuates ROS Production by Activating the Expression of ROS-Scavenging Genes during the Differentiation of 3T3-L1 Cells

During the differentiation from preadipocyte to adipocyte, ROS are deeply related to stimulating the early stages of adipocyte differentiation [15]. Therefore, we evaluated the ROS levels using DCFH-DA. Figure 5A shows that ROS levels increased during the differentiation from 3T3-L1 preadipocyte to adipocyte. Moreover, 20, 50 and 100 µM of fraxetin significantly attenuated the increase in ROS production induced by MDI induction by 45%, 39%, and 28% (all* p *< 0.001), respectively (Figure 5A). These observations are further supported by the transcriptional analysis of scavenging genes, including superoxide dismutase 1 (Sod1) and superoxide dismutase 2 (Sod2). The transcriptional expression levels of both genes were reduced by MDI induction compared with the ND. However, fraxetin restored the Sod1 expression up to approximately two-fold *(p *< 0.001) and elevated the Sod2 mRNA levels up to 2.3-fold *(p *< 0.001) (Figure 5B). Therefore, these results suggest that fraxetin alleviates ROS levels induced by MDI induction during adipocyte differentiation through the activation of antioxidant genes.

### 3.6. Fraxetin Suppresses Adipogenesis via the Regulation of MAPK Signaling

To investigate the MAPK signaling pathway, which is correlated with the early stage of adipogenesis, we evaluated phosphorylation levels of p38, Erk1/2, and Jnk by Western blotting. Phosphorylated p38 was slightly but not significantly reduced in MDI-treated cells compared to the ND cells. However, 20 and 50 µM of fraxetin increased its phosphorylation level by 1.34-fold and 1.49-fold (both* p *< 0.05), respectively (Figure 6A). However, phosphorylated Erk1/2 and Jnk were increased significantly by MDI induction compared to ND cells. In response to 100 µM of fraxetin, the protein expression of phosphorylated Erk1/2 was reduced to 0.50-fold and Jnk to 0.69-fold (both* p *< 0.01) (Figure 6B,C). Our observations indicate that fraxetin suppresses the early phase of adipogenesis through the activation of p38 and the inhibition of Erk1/2 and Jnk.

### 3.7. Effect of the Combination of Fraxetin and MAPK Inhibitors on Lipid Accumulation

To determine the influence of the MAPK signaling pathway on lipid accumulation caused by fraxetin, we performed oil red O staining with the combination of fraxetin and the following three MAPK-specific inhibitors: SB203580, U0126, and SP600125, which were assigned to hinder p38, Erk, and Jnk, respectively (Figure 7A). Compared to the cells treated with only fraxetin, lipid droplets stained by oil red O solution were significantly increased by SB203580 treatment from 60 to 87% *(p *< 0.01). Conversely, compared to the fraxetin-treated groups, U0126 lowered lipid accumulation by 49% *(p *< 0.05). However, SP600125 did not alter the effect of fraxetin on lipid accumulation.

## 4. Discussion

Recent studies have shown that obesity and overweight are crucial hazardous factors for various diseases, including cardiovascular disease, cancer, and metabolic disease [29,30,31]. Adipogenesis and fat storage in adipocytes is greatly associated with the development of obesity [32]. Therefore, many researchers have tried to develop anti-obesity drugs that regulate adipogenesis and lipid metabolism. The conventional therapeutic drugs for the prevention and treatment of obesity are usually correlated to adverse effects and drug dependence [33]. Natural products are under exploration as an alternative to these problems because they are safe and effective [34]. Fraxetin, a coumarin derived from *Fraxinus rhynchophylla*, has versatile pharmacological properties and rare side effects [35]. Our present study derived three major findings. First, fraxetin suppresses adipogenesis at the early phase in 3T3-L1 cells. Second, by activating the antioxidant-related genes, fraxetin ameliorates ROS levels induced by MDI induction. Third, the inhibitory effect of fraxetin on the formation of lipid droplets was mediated by the MAPK signaling pathway.

Adipogenesis is a major pathway that enlarges the adipose tissue, during which various transcriptional factors are modulated and activated. Among them, PPARγ and C/EBP families are the representative regulators of adipogenesis [36]. Adipogenesis from 3T3-L1 preadipocytes into mature adipocytes involves four steps: growth arrest, mitotic clonal expansion, early-stage differentiation, and terminal differentiation [37]. Several factors organize each step of adipogenesis. Among all stages, the early stage of adipocyte differentiation is important because C/EBPβ and C/EBPδ, which are actively expressed in the early stage, subsequently stimulate C/EBPα and its partner PPARγ [38,39]. The elevated activities of the C/EBP family and PPARγ are usually implicated in the activation of several lipogenesis-related genes, including *Fabp4*, *Fasn*, and *Srebf1* [40]. *Fasn* is an essential enzyme for the biosynthesis of long-chain fatty acids [41]. *Fabp4*, commonly known as adipocyte protein 2 (aP2), has key roles in lipid homeostasis and fatty acid transport [42]. *Srebf1* plays an important role as a bridge between C/EBP signaling and the adipogenesis pathway [43]. Several studies have reported that some natural products exert anti-obesity effects by inhibiting these lipogenesis-related genes [44,45]. Various natural products exert direct or indirect antiadipogenic effects by suppressing themselves or their up/downstream modulators. Apigenin, a flavonoid, suppresses adipocyte differentiation by downregulating PPARγ and C/EBPα [33]. Isoflavonoids from *Crotalaria albida* exert inhibitory effects on adipogenesis as a PPARγ antagonist [46]. Extracts of copperleaf ameliorate obesity by reducing PPARγ and C/EBPα expression in a high-fat diet mouse model [47]. Consistent with these reports, we revealed that fraxetin reduced protein and transcriptional levels of C/EBPα and PPARγ. Our study showed that fraxetin treatment exerts inhibitory effects on lipid accumulation regardless of treatment period. However, the inhibitory effect was most marked during day 0–2 of the treatment than that of other periods. Fraxetin also suppressed mRNA levels of *Klf5*, which is essential for differentiation in the early stage of adipogenesis [48]. Fraxetin alleviated the increased levels of adipocyte-differentiation-related genes induced by MDI induction, such as *Fabp4*, *Fasn*, and *Srebf1*. Our findings indicate that fraxetin hinders the adipogenesis at an early stage by downregulating PPARγ and C/EBPα expression levels.

Increasing ROS production induced by fat accumulation promotes oxidative stress in blood. Growing evidence has shown that oxidative stress due to the activation of ROS is greatly related to metabolic disorders, including obesity and diabetes [16,49]. In such patients, adipocytes activate NADPH oxidase and inhibit antioxidant enzymes, such as SODs, catalase, and glutathione peroxidase (GPx) [16,50]. SODs, along with GPx and catalase, neutralize the damage from ROS. ROS generation and fat accumulation during adipogenesis are regulated by the C/EBP family and PPARγ [51]. Therefore, the suppression of ROS production during adipogenesis is a potential strategy for ameliorating obesity-related metabolic diseases. Our study results show that fraxetin impedes ROS production induced by MDI induction during adipogenesis. This was further confirmed by evaluating the mRNA expression of ROS-scavenging genes, *Sod1* and *Sod2*. The fraxetin-containing MDI induction medium increased the transcriptional expression of antioxidant enzymes. Our observations suggest that fraxetin has antioxidant properties and exerts an inhibitory effect on adipocyte differentiation.

MAPK pathway regulation is important for adipogenesis [52]. It is subdivided into three pathways: p38, extracellular signal-regulated kinases (ERKs), and c-Jun amino-terminal kinases (JNKs) MAPK [53]. Several reports suggest that MAPK pathways play an important role in modulating adipocyte differentiation. ERK signaling pathways have important roles in regulating adipogenesis, as the inhibition of ERK1/2 phosphorylation exerts an inhibitory effect on adipocyte differentiation. The roles of ERK1/2 in adipogenesis were verified with ERK activator FGF-2 and ERK inhibitor PD98059 [54]. ERK phosphorylation is crucial for activating C/EBPα and PPARγ [55,56]. The function of p38 MAPK in adipogenesis is still controversial [52,57,58,59]. Several studies have reported that p38 MAPK pathways can negatively affect adipogenesis. Aouadi. M et al. have suggested that p38 was more activated in preadipocytes than in adipocytes of obese mice, suggesting that p38 MAPK has an inhibitory role in adipocyte differentiation [52]. Conversely, evidence suggests that p38 MAPK has a role in promoting adipogenesis [58,59]. These contradictory functions of p38 MAPK in adipocyte differentiation are probably attributed to the versatility of p38 MAPK in interacting with various intracellular molecules and environments. Our data showed that fraxetin exerts inhibitory effects on adipogenesis by the downregulation of Erk1/2 and Jnk and the upregulation of p38 MAPK. These results were further verified by using several MAPK inhibitors, including SB203580, U0126, and SP600125, through oil red O staining. Various studies have demonstrated that the MAPK pathway is also closely related to the regulation of ROS signaling. ROS accumulation can stimulate the activation of MAPK signaling pathways [60]. The suppression of ROS production by antioxidants obstructs MAPK activation, demonstrating that ROS are involved in MAPK activation [61]. Our observations indicated that fraxetin simultaneously suppressed ROS accumulation and ERK1/2 MAPK in 3T3-L1, suggesting that fraxetin may regulate adipogenesis through MAPK signaling driven by ROS. Further study is required on whether the inhibition of ROS production caused by fraxetin is subjected to the MAPK signaling pathway in 3T3-L1.

## 5. Conclusions

In conclusion, our study revealed the underlying mechanisms by which fraxetin exerts an anti-adipogenesis effect during the early stage of adipogenesis (Figure 8). We confirmed that fraxetin ameliorates elevated ROS production by activating ROS-scavenging genes. Furthermore, these effects were mediated by MAPK signaling pathways. Therefore, fraxetin can be a potential anti-obesity agent. Our limitation is that we cannot solve the problem of whether the inhibitory effect of fraxetin on adipogenesis affects the person who is in positive energy balance. Although our findings are limited in vitro, this study might serve as the basis for further in vivo studies and the development of an innovative agent for treating obesity.

## Figures and Tables

**Figure 1 antioxidants-11-01893-f001:**
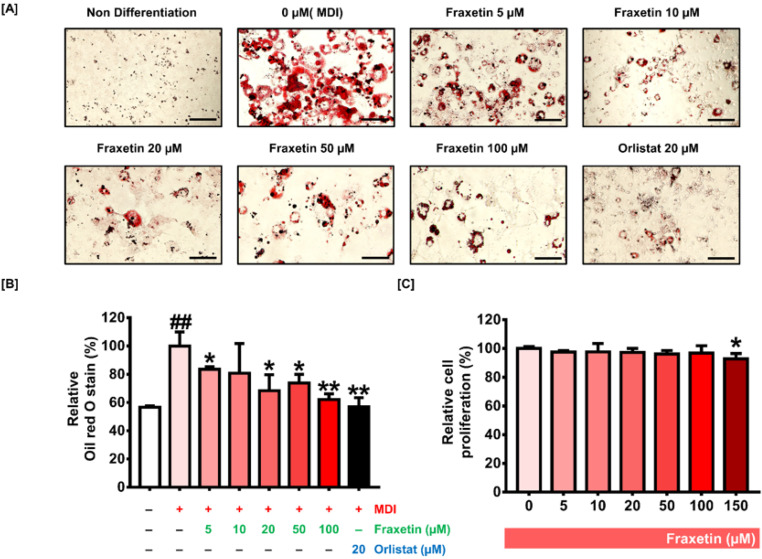
Impact of fraxetin on the formation of lipid droplets and proliferation of 3T3-L1 cells. (**A**,**B**) Nondifferentiated cells and mature adipocytes were incubated with different doses of fraxetin (0, 5, 10, 20, 50, and 100 µM). Relative lipid accumulation stained by oil red O was measured by a microplate reader at 490 nm of optical density. Orlistat (20 µM) was treated as a positive control. Scale bar: 50 µm. The different colors in bar graph represent the different concentration of fraxetin. Black bar indicates orlistat. (**C**) Cell proliferation analysis was conducted in 3T3-L1 preadipocytes. All data were obtained in triplicate. The symbols “*” and “**” represent significant differences between the differentiated group and others (** p *< 0.05, *** p *< 0.01). “##” means considerable differences between the nondifferentiated group and others (##* p *< 0.01), as verified using one-way analysis of variance (ANOVA), followed by Dunnett’s post hoc analysis.

**Figure 2 antioxidants-11-01893-f002:**
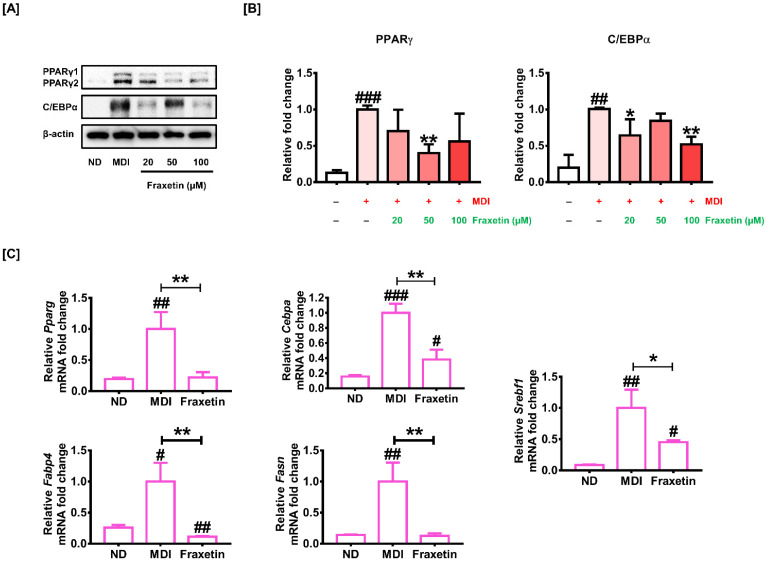
Effect of fraxetin on the expression levels of adipogenesis-related proteins and genes in 3T3-L1 cells. (**A**,**B**) Protein expression levels of PPARγ and C/EBPα were assessed with Western blot analysis. The 3T3-L1 cells were incubated with or without fraxetin (20, 50, and 100 µM) for differentiation. Then, the cells were lysed, and a Western blot was performed. The protein levels of PPARγ and C/EBPα were normalized with β-actin. The different colors in bar graph represent the different concentration of fraxetin. (**C**) The transcriptional expressions of Pparg, Cebpa, Fabp4, Fasn, and Srebf1 genes were determined by RT-qPCR. GAPDH was used for the normalization of each gene expression. All experiments were performed at least three times. The asterisk symbol represents significant differences between the differentiated group and others (** p *< 0.05, *** p *< 0.01). The crosshatch mark represents significant differences between the nondifferentiated group and others (#* p *< 0.05, ##* p *< 0.01, ###* p *< 0.001), as verified using unpaired t-tests or one-way analysis of variance (ANOVA), followed by Dunnett’s post hoc analysis.

**Figure 3 antioxidants-11-01893-f003:**
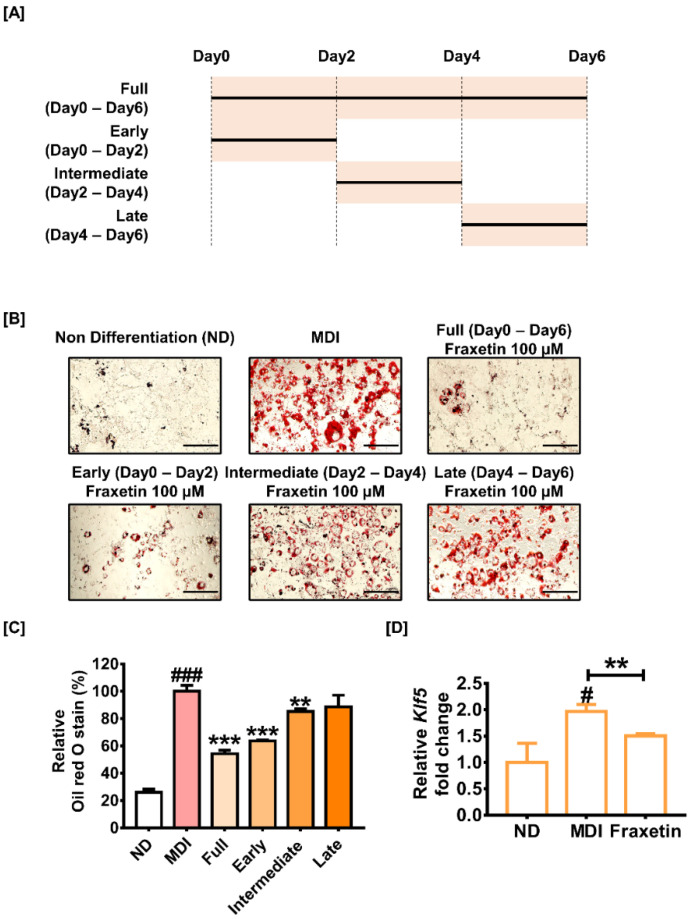
Effect of fraxetin on various stages of adipogenesis in 3T3-L1 cells. (**A**) Schematic diagram of the treatment period of fraxetin (100 µM) in confluent 3T3-L1 cells during adipogenesis. (**B**,**C**) The nondifferentiated cell and differentiated cells treated with fraxetin (100 µM) at different periods of adipogenesis were stained with oil red O solution on day 8. Relative lipid droplets eluted by isopropanol were quantified by measuring the absorbance at 490 nm. Scale bar: 50 µm. (**D**) Transcriptional expression of the *Klf5* gene was analyzed by RT-qPCR. All data were obtained in triplicate. The asterisk symbol represents significant differences between the MDI-treated group and others (*** p *< 0.01, **** p *< 0.001). The crosshatch symbol represents significant differences between the nondifferentiated group and others (#* p *< 0.05, ###* p *< 0.001), as verified using unpaired *t*-tests or one-way analysis of variance (ANOVA), followed by Dunnett’s post hoc analysis.

**Figure 4 antioxidants-11-01893-f004:**
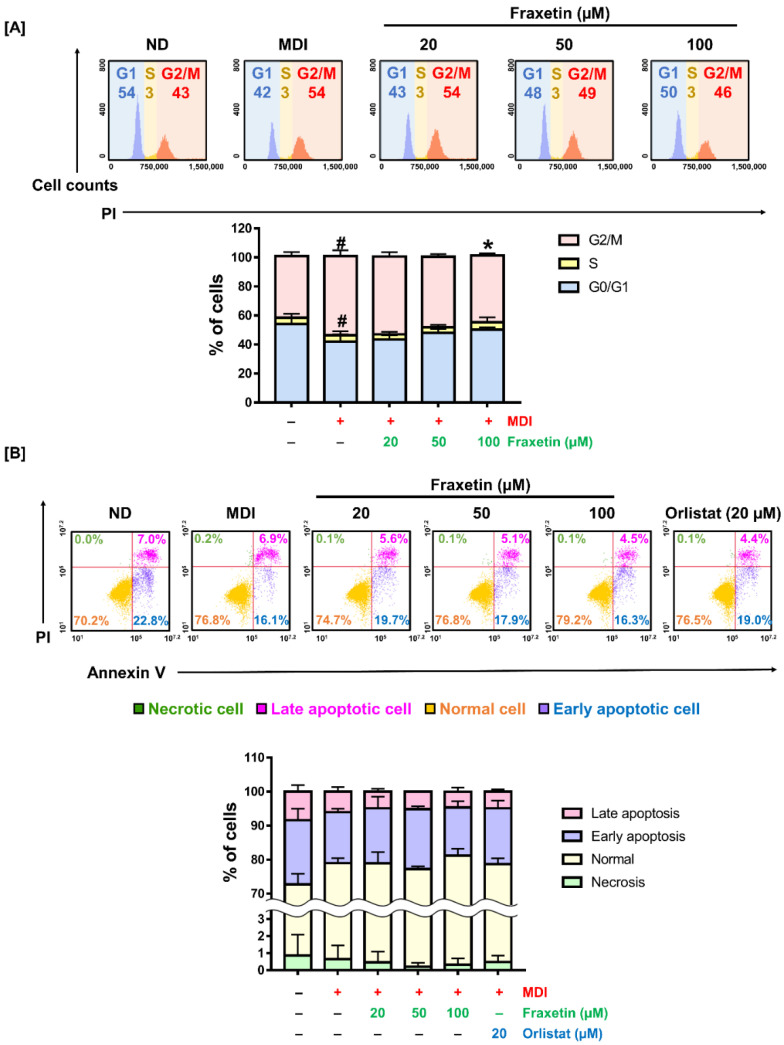
Fraxetin triggers cell-cycle shifts without apoptotic cell death. (**A**) Preadipocytes were incubated with diverse doses of fraxetin in an MDI induction medium for 2 days. After fixation, cells were stained by PI-retaining ribonuclease A and evaluated using flow cytometry. Each value in the histogram was determined by counting 10,000 events. The ratio of cells at each phase was represented in each histogram. (**B**) The impact of fraxetin with respect to apoptosis in MDI-treated cells was evaluated with Annexin V and PI double-staining assay. Confluent preadipocytes were incubated with different concentrations of fraxetin in an MDI induction medium for 2 days. Cells were treated by PI and Annexin V and assessed by flow cytometry. The number of cells in each dot plot was 10,000. All data were obtained in triplicate. The crosshatch mark indicates significant differences among the nondifferentiated group and others (#* p *< 0.05). The asterisk symbol represents significant differences between the MDI-treated group and others (** p *< 0.05), as verified using one-way analysis of variance (ANOVA), followed by Dunnett’s post hoc analysis.

**Figure 5 antioxidants-11-01893-f005:**
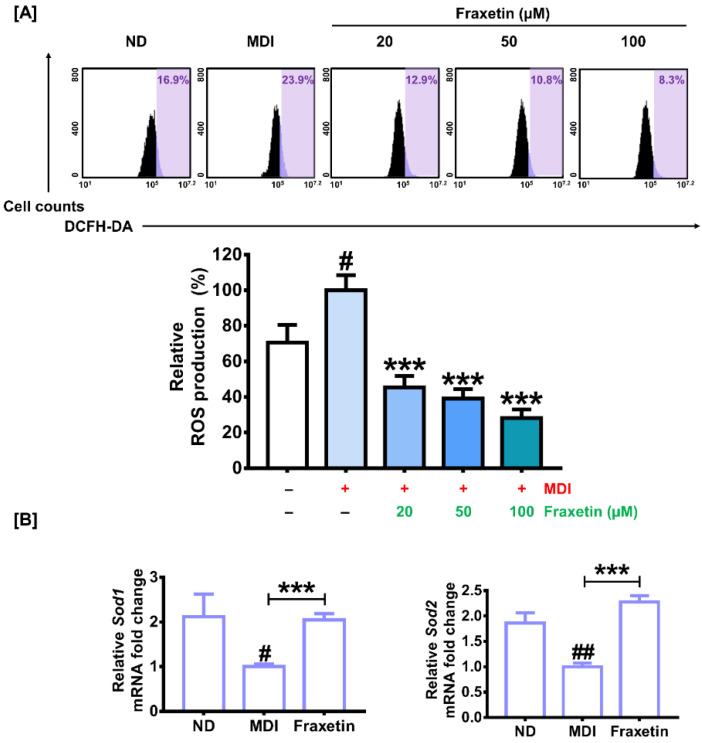
Effect of fraxetin on ROS production and the transcriptional expression of scavenging genes during the adipogenesis of 3T3-L1 cells. (**A**) ROS production was evaluated with DCFH-DA using flow cytometry. Relative ROS production is represented as purple shadow in a bar graph. The different colors in bar graph represent the different concentration of fraxetin. (**B**) The mRNA expressions of scavenging genes (*Sod1*, *Sod2*) were analyzed with RT-qPCR. Each gene was normalized with the GAPDH gene. All data were obtained in triplicate. The asterisk indicates significant differences between the MDI-treated group and others (**** p *< 0.001). The crosshatch symbol represents significant differences between the nondifferentiated group and others (#* p *< 0.05, ##* p *< 0.01), as verified using unpaired *t*-tests or one-way analysis of variance (ANOVA), followed by Dunnett’s post hoc analysis.

**Figure 6 antioxidants-11-01893-f006:**
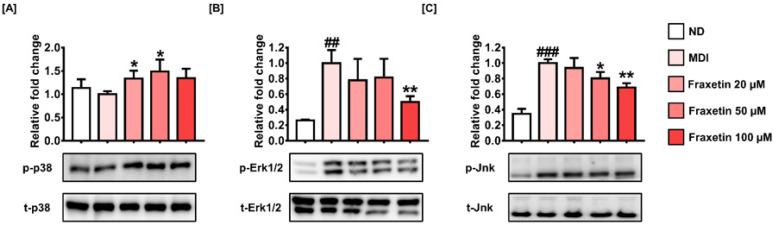
Effect of fraxetin on the regulation of adipogenesis through MAPK signaling. The protein expressions of the MAPK signaling pathway, including (**A**) p38, (**B**) Erk1/2, and (**C**) Jnk phosphorylation, were analyzed with Western blotting. Each phosphorylation level was normalized with each total protein level. All data were obtained in triplicate. The asterisk symbol represents significant differences between the MDI-treated group and others (** p *< 0.05, *** p *< 0.01). The crosshatch symbol represents meaningful differences among the nondifferentiated group and others (##* p *< 0.01, ###* p *< 0.001), as verified using one-way analysis of variance (ANOVA), followed by Dunnett’s post hoc analysis.

**Figure 7 antioxidants-11-01893-f007:**
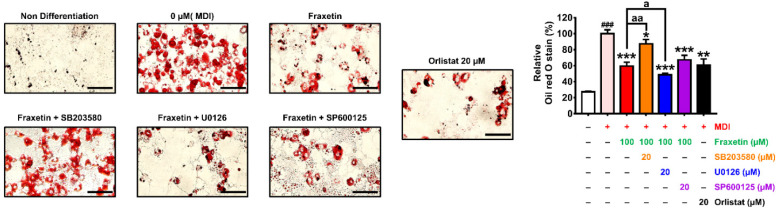
Effect of the combination of fraxetin and MAPK inhibitors on lipid accumulation. Fraxetin was cotreated with MAPK signaling inhibitors including SB203580, U0126, and SP600125 in the presence of an MDI induction medium. Then, the formation of lipid droplets was evaluated with oil red O solution. Relative staining rates were assessed by measuring the optical density at 490 nm. Scale bar: 50 µm. The red and black color represent the treatment with fraxetin and orlistat in the presence of MDI medium, respectively. The orange, blue, and purple colors in bar graph indicate the treatment with SB203580, U0126, and SP600125 in the presence of fraxetin, respectively. Experiments were performed at least three times. The crosshatch represents significant differences between the nondifferentiated group and others (###* p *< 0.001). The asterisk symbol represents significant differences between the MDI-treated group and others (** p *< 0.05, *** p *< 0.01, **** p *< 0.001). The symbol “a” *(p *< 0.05) and “aa” *(p *< 0.01) represent the significant effect of combination treatment between the fraxetin and MAPK-inhibitor-treated groups, as verified using one-way analysis of variance (ANOVA), followed by Dunnett’s post hoc analysis.

**Figure 8 antioxidants-11-01893-f008:**
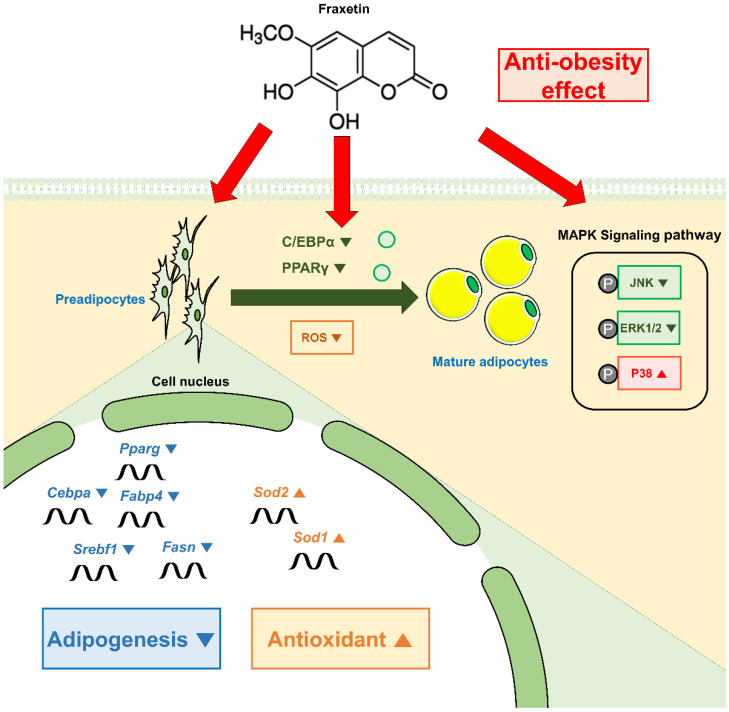
Schematic illustration of fraxetin on 3T3-L1 cells.

**Table 1 antioxidants-11-01893-t001:** The detailed information about the antibodies we used.

Chemicals	Catalog Number	Supplier
p-JNK (Thr183/Tyr185)	4668	Cell Signaling Technology (Beverly, MA, USA)
JNK	9252	Cell Signaling Technology
p-ERK1/2 (Thr202/Tyr204)	9101	Cell Signaling Technology
ERK1/2	4695	Cell Signaling Technology
p-P38 (Thr180/Tyr182)	4511	Cell Signaling Technology
P38	9212	Cell Signaling Technology
PPARγ	2443	Cell Signaling Technology
C/EBPα	8178	Cell Signaling Technology
β-actin	sc-47778	Santa Cruz Biotechnology (CA, USA)
SB203580	BML-EI286	Enzo Life Sciences (Farmingdale, NY)
U0126	BML-EI282	Enzo Life Sciences
SP600125	BML-EI305	Enzo Life Sciences

**Table 2 antioxidants-11-01893-t002:** The primers we used in qPCR.

Gene	Size (bp)	GenBank Accession No.	Primer Sequence (5’→3’)
Peroxisome proliferator-activated receptor gamma (Pparg)	126	NM_001127330.2	F: CGAGTCTGTGGGGATAAAGC
			R: CCGGCAGTTAAGATCACACC
CCAAT enhancer binding protein alpha (Cebpa)	87	NM_001287514.1	F: GGTGGACAAGAACAGCAACG
			R: CGTTGTTTGGCTTTATCTCG
Fatty-acid-binding protein 4 (Fabp4)	131	NM_024406.3	F: GCCCAACATGATCATCAGC
			R: TCACCTTCCTGTCGTCTGC
Fatty acid synthase (Fasn)	95	NM_007988.3	F: AGCACACATCCTAGGCATCC
			R: GAACTTCCACACCCATGAGC
Sterol regulatory-element-binding transcription factor 1 (Srebf1)	146	NM_001313979.1	F: AGGTCACCGTTTCTTTGTGG
			R: AGTTCAACGCTCGCTCTAGG
Kruppel-like factor 5 (Klf5)	127	NM_009769.4	F: ACAAATCCCAGAGACCATGC
			R: CTAGTGAACTCGGGGAGAGC
Superoxide dismutase 1 (Sod1)	87	NM_011434.2	F: CAAGATGACTTGGGCAAAGG
			R: AATCCCAATCACTCCACAGG
Superoxide dismutase 2 (Sod2)	81	NM_013671.3	F: GTGTCTGTGGGAGTCCAAGG
			R: GCAGGCAGCAATCTGTAAGC
Glyceraldehyde-3-phosphate dehydrogenase (GAPDH)	130	NM_001289726.1	F: AACTTTGGCATTGTGGAAGG
			R: ATGCAGGGATGATGTTCTGG

## Data Availability

The data are contained within this article.

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
