# Peer review of "Suppressive Effect of Fraxetin on Adipogenesis and Reactive Oxygen Species Production in 3T3-L1 Cells by Regulating MAPK Signaling Pathways"

_antioxidants, 2022, doi:10.3390/antiox11101893_

Round 1

Reviewer 1 Report

In the present study, the authors investigate the effectiveness of fraxetin, a plant derived, cumarin-like extract, in preventing or mitigating adipogenesis, with th eultimate intent of determining whether the product can be beneficial in treatment or amelioration of obesity. For their study, the authors used 3T3-L1 cells cultured and differentiated in vitro as adipocytes, and tested the dose-dependent effectiveness of fraxetin at various stages of the adipogenic differentiation process.

The results reported in the study indicate that fraxetin exerts a dose-dependent inhibition on adipogenesis, down-regulating C/EBP alpha and PPAR-gamma and decreasing the formation of lipid droplets in 3T3-L1 cells. Fraxetin also decreases ROS production. Further, the cells present with normal or slightly higher than normal SOD1 and SOD2 respectively. Fraxetin also increased the number of cells in the Go/G1 phase, while decreased the number of cells in the G2/M phase. Interestingly, the effect of fraxetin was more pronounced in the initial phase of adipogenesis differentiation. Lastly, the effect of fraxetin appears to occur through a downregulation of phosphorylated ERK1/2 and JNK and an upregulation of phosphorylated p38. 

Comments:

The study appears to be properly conducted and can be followed properly for the most part. The main comment about the study is that the authors present the effect of fraxetin on three different signaling pathways (C/EBP-PPAR-gamma; ROS, and MAPKs) but it is not clear whether a sequential relationship exists among these pathways. While it can be appreciated to some extent that altered regulation of MAPKs can impact cell cycle progression and adipogenesis, less clear is whether a causative relation exists between ROS and adipogenesis or between ROS and MAPKs. It would be helpful if the authors could provide some connection, even at the speculative level, between these modified regulatory mechanisms in the Discussion section, 

Minor

some sentences are difficult to understand: e.g. lines 16-18, 66-67 (what fatal refers to?); 340-342;

line 247: treatment with fraxetin instead of treatment of fraxetin

line 362; replace via with by (...by inhibiting...) 

Author Response

Reviewer 1:

Comments and Suggestions for Authors

In the present study, the authors investigate the effectiveness of fraxetin, a plant derived, cumarin-like extract, in preventing or mitigating adipogenesis, with the ultimate intent of determining whether the product can be beneficial in treatment or amelioration of obesity. For their study, the authors used 3T3-L1 cells cultured and differentiated in vitro as adipocytes, and tested the dose-dependent effectiveness of fraxetin at various stages of the adipogenic differentiation process.

The results reported in the study indicate that fraxetin exerts a dose-dependent inhibition on adipogenesis, down-regulating C/EBP alpha and PPAR-gamma and decreasing the formation of lipid droplets in 3T3-L1 cells. Fraxetin also decreases ROS production. Further, the cells present with normal or slightly higher than normal SOD1 and SOD2 respectively. Fraxetin also increased the number of cells in the Go/G1 phase, while decreased the number of cells in the G2/M phase. Interestingly, the effect of fraxetin was more pronounced in the initial phase of adipogenesis differentiation. Lastly, the effect of fraxetin appears to occur through a downregulation of phosphorylated ERK1/2 and JNK and an upregulation of phosphorylated p38.

Response: We appreciate the reviewer’s valuable comments and suggestions on our manuscript. We have substantially revised our manuscript according to the reviewer’s suggestions. To address the reviewer’s comments, we prepared a point-by-point response to each comment of the reviewer and highlighted changes in text of the manuscript in yellow.

Comments:

The study appears to be properly conducted and can be followed properly for the most part. The main comment about the study is that the authors present the effect of fraxetin on three different signaling pathways (C/EBP-PPAR-gamma; ROS, and MAPKs) but it is not clear whether a sequential relationship exists among these pathways. While it can be appreciated to some extent that altered regulation of MAPKs can impact cell cycle progression and adipogenesis, less clear is whether a causative relation exists between ROS and adipogenesis or between ROS and MAPKs. It would be helpful if the authors could provide some connection, even at the speculative level, between these modified regulatory mechanisms in the Discussion section,

Response: We appreciate the reviewer’s valuable comments and totally agree with you. According to the reviewer’s suggestion, we add some relationship between ROS production and MAPK in 3T3-L1 at the speculative level at the end of Discussion section.-

Minor

some sentences are difficult to understand: e.g. lines 16-18, 66-67 (what fatal refers to?); 340-342;

Response: We appreciate the reviewer’s valuable comments. According to reviewers’ suggestions, we revised our manuscript more precisely.

line 247: treatment with fraxetin instead of treatment of fraxetin

Response: We appreciate the reviewer’s valuable comments. According to reviewers’ suggestions, we changed ‘treatment of fraxetin’ into ‘treatment with fraxetin’ in line 247.

line 362; replace via with by (...by inhibiting...)

Response: We appreciate the reviewer’s valuable comments. According to reviewers’ suggestions, we replaced ‘via’ with ‘by’ in line 362.

Reviewer 2 Report

This is one of the best papers I've ever reviewed. Great work, congratulations! I've got only one tiny comment: something went wrong with placing figure 1, it's in the middle of the text and the figure caption is two lines below the text. After this correction the paper should be published

Author Response

Reviewer 2:

Comments and Suggestions for Authors

This is one of the best papers I've ever reviewed. Great work, congratulations! I've got only one tiny comment: something went wrong with placing figure 1, it's in the middle of the text and the figure caption is two lines below the text. After this correction the paper should be published

Response: We appreciate the reviewer’s valuable comments and suggestions on our manuscript. According to the reviewer’s opinion, we adjusted the position of Figure 1.

Reviewer 3 Report

Introduction- I think the manuscript would benefit from a more clearly stated overall goal and hypothesis at the end of the introductory segment.

Throughout the paper: I've never heard the term “antiobestic” before.

Line 66- I would suggest replacing the word “fatal”. 

Section 2.10. Please clarify which data were analyzed by t-test and which were analyzed by ANOVA.

The authors should explain why orlistat was used as a positive control. Orlistat inhibits gastric and pancreatic lipase. What effect is it expected to have in 3T3-L1 adipocytes? 

Could the authors show protein expression for FABP4, FASN, SREBF1, SOD1, and SOD2? Why was gene expression chosen for these markers instead of protein? 

The authors should comment on the fact that in positive energy balance, excess energy is stored as triglycerides, most safely within the adipose tissue. Adipocytes expand in two ways to accommodate the excess energy: hypertrophy and hyperplasia. While hyperplasia does increase the number of adipocytes, these adipocytes are generally "healthier" than the adipocytes that have undergone hypertrophy that become insulin resistant and inflammatory. Is it really preferable to inhibit adipocyte differentiation if a person is in a positive energy balance? With limited new adipocyte formation, the excess energy will either be stored in hypertrophied adipocytes or in non-adipocytes. Both have been linked to deleterious effects of obesity on metabolism. 

Author Response

Reviewer 3:

Comments and Suggestions for Authors

Introduction- I think the manuscript would benefit from a more clearly stated overall goal and hypothesis at the end of the introductory segment.

Response: We appreciate the reviewer’s valuable comments and suggestions on our manuscript. According to the reviewer’s suggestion, we clearly stated the goal of our present study at the end of the introduction part and highlighted changes in text of the manuscript in yellow.

Throughout the paper: I've never heard the term “antiobestic” before.

Response: We appreciate the reviewer’s valuable comments. According to reviewers’ suggestions, we replaced ‘antiobestic’ with ‘anti-obesity’ throughout the paper.

Line 66- I would suggest replacing the word “fatal”.

Response: We appreciate the reviewer’s valuable comments. According to reviewers’ suggestions, we replaced ‘fatal’ with ‘hazardous’ in line 65.

Section 2.10. Please clarify which data were analyzed by t-test and which were analyzed by ANOVA.

Response: We appreciate the reviewer’s valuable comments. According to reviewers’ suggestions, we clarified the statistical information in each figure legend.

The authors should explain why orlistat was used as a positive control. Orlistat inhibits gastric and pancreatic lipase. What effect is it expected to have in 3T3-L1 adipocytes?

Response: We appreciate the reviewer’s valuable comments. As reviewer mentioned, orlistat inhibits gastric and pancreatic lipase leading to reduced absorption of dietary fat in the gastrointestinal tract [1]. Several studies have reported that orlistat also inhibited lipid accumulation in 3T3-L1 cells [2, 3]. For this reason, numerous studies have suggested that orlistat can be used as a positive control for in vitro experiment with 3T3-L1.

Could the authors show protein expression for FABP4, FASN, SREBF1, SOD1, and SOD2? Why was gene expression chosen for these markers instead of protein?

Response: We appreciate the reviewer’s valuable comments and totally agree. We just proposed the gene expression of FABP4, FASN, SREBF1, SOD1, and SOD2. That is clearly our limitation. We deeply agree with you, but we are very sorry that it cannot be implemented because of the short deadline. We need more than 2 months to purchase chemicals and perform those experiments. Please understand our circumstances. Instead, we tried to explain the relationship between transcriptional levels of above genes and lipid accumulation or ROS production. Frist, Y. Park and colleague have suggested that anti-obesity effect of plant extraction was shown by confirming the gene expression of FASN, PPARγ, C/EBPα, and SREBF1 in 3T3-L1 cells [4]. Moreover, gene expressions of PPARγ, ACC, and FASN in 3T3-L1 cells were also evaluated in the study of anti-obesity medication [5]. Furthermore, gene expressions of SOD1 and SOD2 were also determined in 3T3-L1 [6]. Although there have been many reports regarding evaluation of the transcriptional expression of lipogenic genes and antioxidant genes in 3T3-L1, further study is required. In a follow-up study, we will plan experiments evaluating the protein levels of various targets and conducting in vivo study.

The authors should comment on the fact that in positive energy balance, excess energy is stored as triglycerides, most safely within the adipose tissue. Adipocytes expand in two ways to accommodate the excess energy: hypertrophy and hyperplasia. While hyperplasia does increase the number of adipocytes, these adipocytes are generally "healthier" than the adipocytes that have undergone hypertrophy that become insulin resistant and inflammatory. Is it really preferable to inhibit adipocyte differentiation if a person is in a positive energy balance? With limited new adipocyte formation, the excess energy will either be stored in hypertrophied adipocytes or in non-adipocytes. Both have been linked to deleterious effects of obesity on metabolism.

Response: We appreciate the reviewer’s valuable comments and totally agree with reviewer’s opinion. That is exactly our limitation. We added our limitation in conclusion section. In a follow-up in vivo study, we will confirm whether inhibition of adipogenesis affects human and mice in positive energy balance.

  1. Birari, R.B. and K.K. Bhutani, Pancreatic lipase inhibitors from natural sources: unexplored potential. Drug Discov Today, 2007. 12(19-20): p. 879-89.
  2. Park, C.H., et al., Inhibition of preadipocyte differentiation and lipid accumulation by 7-O-galloyl-d-sedoheptulose treatment in 3T3-L1 adipocytes. Biomedicine & Preventive Nutrition, 2013. 3(4): p. 319-324.
  3. Hum Park, C., et al., Effectiveness of Chinese prescription Kangen-karyu for dyslipi-demia, using 3T3-L1 adipocytes and type 2 diabetic mice. Drug Discov Ther, 2014. 8(3): p. 121-31.
  4. Park, Y.H., et al., Antiobesity effect of ethanolic extract of Ramulus mori in differentiated 3T3-L1 adipocytes and high-fat diet-induced obese mice. J Ethnopharmacol, 2020. 251: p. 112542.
  5. Tsai, M.C., et al., Effect of Astaxanthin on the Inhibition of Lipid Accumulation in 3T3-L1 Adipocytes via Modulation of Lipogenesis and Fatty Acid Transport Pathways. Molecules, 2020. 25(16).
  6. Nishitani, S., et al., Ketone body 3-hydroxybutyrate enhances adipocyte function. Sci Rep, 2022. 12(1): p. 10080.